# TREE OF OPTIONS: TEMPORALLY EXTENDED WORLD MODELING, PLANNING, AND EXECUTION WITH LARGE LANGUAGE MODELS

## ABSTRACT

With commonsense knowledge embedded, Large Language Models (LLMs) have been repurposed as world models that can be exploited by principled planning algorithms such as Monte Carlo Tree Search (MCTS). Prior works have been limited to exploiting LLMs for low-level world modeling, i.e., predicting immediate next world states and rewards upon primitive actions, which makes them unfit for long-horizon tasks where prediction errors compound quickly over time. This work develops an alternative framework where LLMs perform world modeling on *temporally extended actions* (options), to overcome their limitations in precise world modeling at small temporal scales. At this temporal abstraction level, LLMs will also be competent in suggesting reasonable options, enabling effective planning using MCTS. To execute the planned options with the primitive actions, we again turn to LLMs by prompting them to synthesize code implementing option-conditioned policies, which LLMs are known to excel at. Empirical results in Minecraft show that this approach substantially improves performance over prior LLM-based planners on long-horizon, compositional tasks for embodied agents.

## 1 INTRODUCTION

Pre-trained large language models (LLMs) have shown remarkable effectiveness for solving embodied decision-making tasks like games and robotics. The majority of prior works have applied LLMs as high-level planners that decomposes the task of interest into a series of subgoals (Ahn et al., 2022; Huang et al., 2022b;a) and for generating code that implements low-level controllers over primitive actions such as motor commands for robots (Liang et al., 2022; Singh et al., 2022). In a minority of prior works (Hao et al., 2023; Zhao et al., 2023), LLMs have been repurposed as *world models* that predict and evaluate the long-term outcomes of a given sequence of actions on world states (e.g., environment configurations, agent capabilities), on top of which principled model-based methods such as tree search planning can be employed to identify promising actions. While LLM-based world models seem appealing, prior works have restrictively shown their effectiveness in short-horizon tasks, such as object rearrangement (e.g., Blocksworld (Valmeekam et al., 2023), VirtualHome (Puig et al., 2018)) and textual question answering (e.g., math reasoning (Cobbe et al., 2021), logical reasoning Saparov & He (2023)), which require a small number of actions or reasoning steps to complete. This is because these methods model the world upon primitive actions at every tick of the environment and the prediction errors that are inevitable by LLMs compound quickly over time, making them unreliable for long-horizon, complex tasks.

Addressing the challenge, we propose a novel paradigm where LLM-based world models are driven by temporally extended actions, firstly formalized and referred to as *options* by Sutton et al. (1999), which are represented by natural language in this work. With the option-driven world model, we perform tree search that is guided by another LLM prompting that proposes and evaluates candidate options, outputting a sequence of options as subgoals for the task. This effectively shortens the task horizon and greatly alleviates the issue of LLM prediction errors. As a critical design, the tree search in ToO over options incorporates the LLM-based world model to predict state dynamics upon candidate options. As illustrated in Figure 1, this is a key difference of ToO from a naive application of prompting techniques such as Tree of Thoughts (ToT) (Yao et al., 2023) that lacks world modeling, which we find crucial to performance in our experiments. When an option is proposed, it is first

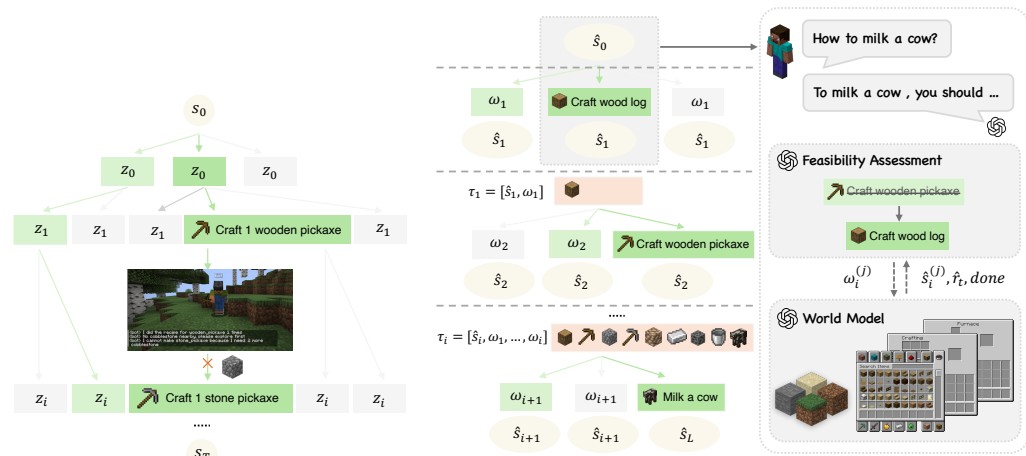

Figure 1: *Left:* A naive Tree-of-Thought (ToT) approach to high-level planning in Minecraft. Each rectangle represents a thought step, while circles represent states of the environment or task. Thought steps are connected by arrows during the reasoning process, and a coherent sequence is generated that represents the intermediate steps in achieving the task goal. *Right:* Proposed options in ToO first pass through a feasibility assessment and are then forwarded to an option-level world model, which predicts the next state, reward, and termination. These transitions expand the search tree and guide the selection of subsequent options.

evaluated by the feasibility module. Only options assessed as feasible are forwarded to the world model, otherwise, they are replaced with the prerequisite steps. The world model then predicts the next state, associated reward, and whether the option would lead to a terminal condition. These predictions are used to expand the search tree. After a sequence of options has been identified, we again leverage LLMs to generate code that implements in-option policies over primitive actions to execute options. The low-level iterative option execution process is illustrated in Figure 5. We refer to this paradigm as Tree of Options (ToO).

By design, ToO leverages LLMs' strengths in commonsense knowledge, high-level planning, and code generation while circumventing their weaknesses in low-level modeling at fine temporal scales. ToO treats LLMs just as black boxes to prompt and no finetuning is required. Even, ToO requires no gradient-based reinforcement learning (RL) as the only trial and error, if any, is performed by iterative code generation (Wang et al., 2023a) for option execution. This makes ToO easy to implement and deploy. Our ToO demonstrates superior performance for several intricate, long-horizon embodied tasks in MineDojo (Fan et al., 2022), an environment build for testing embodied agents in the game of Minecraft.

## 2 BACKGROUND

This section sets up preliminaries for decision-making formalisms and LLM prompting.

**MDPs and options.** The decision-making task faced by our agent can be formulated as a Markov decision process (MDP) that consists of a set of states $\mathcal{S}$, a set of action $\mathcal{A}$, a state transition function $p : \mathcal{S} \times \mathcal{A} \to \Delta(\mathcal{S})$ where $\Delta(\mathcal{X})$ denotes the collection of probability measures on $\mathcal{X}$, and a reward function $r : \mathcal{S} \times \mathcal{A} \to \mathbb{R}$. Starting in some initial state $s_0$ chosen by the task, at each discrete time step $t = 0, 1, \ldots$, the agent fully observes the state $s_t \in \mathcal{S}$, and chooses an action $a_t \in \mathcal{A}$ to take; the next state is sampled from the transition dynamics as $s_{t+1} \sim p(\cdot|s_t, a_t)$ while the agent receives the step reward $r_t := r(s_t, a_t)$; this yields trajectory $(s_0, a_0, r_0, s_1, \ldots)$. A (Markovian stationary) policy specifies an action-select rule as a probability distribution over the action space conditioned on the state, $\pi : \mathcal{S} \to \Delta(\mathcal{A})$. The agent aim to find a policy that maximizes the expected cumulative discounted reward, $\mathbb{E}_\pi[\sum_{t=0}^{\infty} \gamma^t r_t]$, where $\gamma \in [0, 1)$ is the discount factor.

Sutton et al. (1999) formalized the idea of temporally extended actions with the notion of options. An option $\omega \in \Omega$ is specified by a tuple $(\pi_\omega, \beta_\omega)$, where $\pi_\omega : \mathcal{S} \to \Delta(\mathcal{A})$ is its in-option policy, and $\beta_\omega :$

$\mathcal{S} \to \{0, 1\}$ is its termination function. The options are executed in a *call-and-return* manner: the agent executes the current option $\omega$ by following its in-option policy $\pi_\omega$ until termination indicated by termination function $\beta_\omega$, at which point the agent picks the next option in $\Omega$ to execute.

**LLM prompting.** Let $\texttt{LLM}(y|x)$ denote the probability of getting output $y$ with input $x$ given to an LLM with $x, y$ being token sequences. We use $\texttt{LLM}_{\texttt{prompt}}(y|x) = \texttt{LLM}(y \mid \texttt{prompt}(x))$ to denote the practice of wrapping input $x$ with a certain prompt format (e.g., task descriptions, input-output examples) with wrapper $\texttt{prompt}(\cdot)$, which is the most common way to use LLMs. To improve multi-step reasoning capabilities for complex input-output relations, Chain of Thoughts (CoT) by Wei et al. (2022) introduces intermediate reasoning steps between input $x$ and output $y$, enabling the model to generate a series of logically coherent intermediate steps. The thoughts, as token sequences, are sampled sequentially $z_i \sim \texttt{LLM}_{\texttt{CoT}}(\cdot | x, z_1, \ldots, z_{i-1})$. Then the final output $y$ is generated conditioned on the full thought sequence: $y \sim \texttt{LLM}_{\texttt{CoT}}(y \mid x, z_0, \ldots, z_T)$. Building on CoT, Tree of Thoughts (ToT) by Yao et al. (2023) introduces a tree-structured reasoning paradigm by sampling multiple i.i.d. thoughts at each reasoning step: $\{z_i^{(j)}\}_j \overset{\text{iid}}{\sim} \texttt{LLM}_{\texttt{ToT}}(\cdot | \tau_{i-1})$ where $\tau_{i-1} := [x, z_1, \ldots, z_{i-1}]$. In ToT, the path from the root to each tree node represents $\tau_{i-1}$, a trace of intermediate reasoning steps, and an edge denotes a candidate next-step thought $z_i^{(j)}$ that expand the current reasoning trace. Each complete path in the tree corresponds to a full candidate reasoning trajectory. By exploring multiple candidate paths, ToT demonstrates superior performance over CoT in complex tasks such as mathematical reasoning and creative generation.

## 3 TREE OF OPTIONS

This section describes our Tree-of-Options (ToO) paradigm that applies LLM prompting techniques to implement the option framework. As an overview, our ToO solves a decision-making task in the following two phases:

- **Options planning** (Section 3.1). In the first phase, ToO prompts an LLM for the given task to generate a sequence of options represented by natural language. These options are planned out by Monte Carlo Tree Search (MCTS) that is guided by an option-level world model and an option generator, both of which are implemented via LLM prompting techniques.
- **Options execution** (Section 3.2). In the second phase, ToO executes the planned option sequence in the call-and-return manner, with the in-option policies and termination function implemented by LLM-generated code.

In our experiments, the planning is one-shot, i.e., by invoking the MCTS procedure from the initial state, the output sequence of options is used to solve the task without need of additional planning. We find this one-shot planning already demonstrates superior performance on several complex and long-horizon tasks in Minecraft. A straightforward extension is online option-level planning, i.e., re-planning from current states during task execution for an updated sequence of future options.

### 3.1 OPTION-LEVEL WORLD MODELING AND PLANNING WITH LLMS

ToO builds a search tree of options where an edge $\hat{s}_{i-1} \xrightarrow{\omega_i} \hat{s}_i$ represents option $\omega_i$ executed with $\hat{s}_{i-1}$ being its initial state and $\hat{s}_i$ being its terminal state. Here, each tree node stores state $\hat{s}_i$ that is obtained from the *option-driven dynamics predictor* implemented by prompting an LLM:

$$\hat{s}_i \sim \texttt{LLM}_{\texttt{dynamics}}(\hat{s}_i \mid \hat{s}_{i-1}, \omega_i). \tag{1}$$

As the initial state $s_0$ is observable, we set $\hat{s}_0 = s_0$ as the root node. Therefore, a path from the root to a node of $\hat{s}_i$ yields a trajectory $\tau_i := [\hat{s}_i, \omega_1, \ldots, \omega_i]$, with a slight abuse of notation against Section 2. ToO adopts an MCTS-like algorithm to build the tree and search over candidate option sequences to execute, iterating over the following four procedures:

**1. Option selection:** A search iteration begins with a traversal from the root node to a leaf node. We select the edges/options during the traversal by applying the Upper Confidence Bound (UCB) strategy to balance exploration and exploitation:

$$\omega_{i+1} \leftarrow \underset{\omega \in E(\tau_i)}{\arg\max} \left[ Q(\tau_i, \omega) + c \cdot \sqrt{\tfrac{\ln N(\tau_i)}{N(\tau_i, \omega)+1}} \right]$$

where $E(\tau_i)$ is the set of outgoing edges, i.e., candidate next options in the tree, from node $\tau_i$; $Q(\tau_i, \omega)$ is an estimate the expected total reward starting from executing $\omega$ from node $\tau_i$, as detailed in the Backpropagation procedure; $N(\tau_i)$ and $N(\tau_i, \omega)$ denotes the number of times the node $\tau_i$ and the node-option pair $(\tau_i, \omega)$ have been visited, respectively; and the exploration weight $c$ quantifies the exploration and exploitation tradeoff.

**2. Expansion:** Once a leaf node $\tau_{i-1}$ is reached, we expands the tree by proposing multiple candidate next options from the the leaf node that are generated by prompting an LLM:

$$\tilde{\omega}_i^{(1)}, \cdots, \tilde{\omega}_i^{(k)} \sim \text{LLM}_{\text{propose}}\big(\tilde{\omega}_i^{(1)}, \cdots, \tilde{\omega}_i^{(k)} \mid \tau_{i-1}\big) \qquad (2)$$

Here, to improve prompting efficiency, $\text{LLM}_{\text{propose}}$ generates multiple candidate options upon a single prompt.

We have found LLMs easily propose linguistic plausible yet infeasible options. As a remedy, we incorporate a feasibility test that immediately applies to the proposed options before expansion. Specifically, another prompting of LLM is invoked to assess the feasibility of the proposed options in the current state of the node, taking into account task-specific information. For options deemed infeasible, we prompt the LLM to proactively generate semantically aligned prerequisite steps that pursue similar high-level intentions while satisfying execution constraints, returning modified options with better feasibility as the final recommendation:

$$\omega_i^{(j)} \sim \text{LLM}_{\text{feasibility}}\big(\omega_i^{(j)} \mid \tau_{i-1}, \tilde{\omega}_i^{(j)}\big), \quad j = 1, \ldots, k. \qquad (3)$$

We call (2) and (3) together as our *option generator*. In contrast to post-execution feedback mechanisms such as self-validation and self-reflection in recent works , our option generator ensures that the expanded branches are not only linguistically coherent but also environmentally executable.

Conditioned on the recommended options $\omega_i^{(j)}$, we use the dynamic predictor to simulate the state-option transition as in (1), i.e., $\hat{s}_i^{(j)} \sim \text{LLM}_{\text{dynamics}}\big(\hat{s}_i^{(j)} \mid \hat{s}_{i-1}, \omega_i^{(j)}\big)$, which expands the leaf node $\tau_{i-1} = [\hat{s}_{i-1}, \omega_1, \ldots, \omega_{i-1}]$ with children $\tau_i^{(j)} = \big[\hat{s}_i^{(j)}, \omega_1, \ldots, \omega_{i-1}, \omega_i^{(j)}\big]$.

**3. Option-driven rollout:** If state $\hat{s}_l$ of the newly expanded leaf node $\tau_l$ is a terminal state , we skip this rollout procedure. We tell if a state is terminal by prompting an LLM as termination predictor for a binary indictator done:

$$\text{done} \sim \text{LLM}_{\text{termination}}(\text{done} \mid \hat{s}). \qquad (4)$$

Otherwise, to evaluate the non-terminal state, we simulate a state-action trajectory by repeatedly invoking the option-driven dynamics predictor (1) from it, yielding $(\hat{s}_l, \omega_{l+1}, \hat{s}_{l+1}, \omega_{l+2}, \ldots, \hat{s}_L)$ until reaching a terminal state $\hat{s}_L$ or maximum rollout length $L$, where the options therein are chosen by some option rollout poilcy. For simplicity and similar to Hao et al. (2023), in our experiments, we choose the next option from the option generator that maximizes the immediate reward according to a *reward predictor*:

$$\omega_{l'+1} \leftarrow \arg\max_{\omega \in \Omega(\tau_{l'})} \hat{r}(\tau_{l'}, \omega), \quad l' = l, l+1, \ldots, L-1 \qquad (5)$$

where $\tau_{l'} = [\hat{s}_{l'}, \omega_1, \ldots, \omega_{l'}]$ and $\Omega(\tau_{l'})$ is a set of options proposed and modified by $\text{LLM}_{\text{propose}}$ and $\text{LLM}_{\text{feasibility}}$ as described in the Expansion procedure. In our experiments, we implement reward predictor $\hat{r}$ by prompting an LLM on a set of evaluation questions $\mathcal{Q} = \{q\}$, each question assessing a unique aspect of interest, such as feasibility, task relevance, and logical coherence. The final reward is scalarized with weights $c_q$:

$$\hat{r}(\tau, \omega) = \sum_{q \in \mathcal{Q}} c_q r_q(\tau, \omega), \quad r_q(\tau, \omega) \sim \text{LLM}_{\text{reward}}(\hat{r} \mid \tau, \omega, q). \qquad (6)$$

**4. Backpropagation:** The cumulative reward is then propagated back from the leaf to the root, updating the Q-value of each visited node-option pair as

$$Q(\tau_i, \omega_{i+1}) \leftarrow Q(\tau_i, \omega_{i+1}) + \frac{R_i - Q(\tau_i, \omega_{i+1})}{N(\tau_i, \omega_{i+1})} \quad \text{with } R_i = \sum_{j=i}^{L} \gamma^{j-i} \hat{r}(\tau_j, \omega_{j+1}) \qquad i = 1, \ldots, l$$

which will guide the option selection in the next iteration.

By repeatedly iterating over the four procedures, ToO progressively builds up a search tree over options with Q-value estimates. The final sequence of options to recommend is obtained by greedily choosing the options that maximizes the Q-value estimates.

## 3.2 Options execution with LLM-generated code

In its second phase, ToO executes from the initial state the sequence of options $(\omega_i)_{i=1,2...}$ as recommended from the MCTS in the first phase in the call-and-return manner, where the in-option policies $\pi_{\omega_i}$ over primitive actions are implemented by LLM-generated code. As LLM-generated code is prone to errors, we adopt an iterative prompting mechanism similar to prior work Ni et al. (2023); Shinn et al. (2023); Skreta et al. (2023); Wang et al. (2023a). Specifically, suppose we aim to execute option $\omega_i$ starting from state $s_{t_1}$ at timestep $t_1$, we prompt an LLM to generate the first version of the in-option policy:

$$\pi_{\omega_i}^{(1)} \sim \texttt{LLM}_{\texttt{in-option}}(\cdot \mid s_{t_1}, \omega_i)$$

For iteration $m = 1, 2, \ldots$, $\pi_{\omega_i}^{(m)}$ is executed from state $s_{t_m}$ at timestep $t_m$ to some some $s_{t_{m+1}}$ timestep $t_{m+1}$, so that either $s_{t_{m+1}}$ terminates the option, i.e., $\beta_{\omega_i}(s_{t_{m+1}}) = 1$, according the termination function implemented by another LLM prompting:

$$\beta_{\omega_i}(s_{t_{m+1}}) \sim \texttt{LLM}_\beta(\cdot \mid s_{t_{m+1}}, \omega_i) \tag{7}$$

or the execution is erroneous and the in-option policy is refined as

$$\pi_{\omega_i}^{(m+1)} \sim \texttt{LLM}_{\texttt{in-option}}(\cdot \mid s_{t_{m+1}}, \omega_i, \text{feedback}^{(m)}) \tag{8}$$

where $\text{feedback}^{(m)}$ is the error messages from the code interpreter and/or the task environment when executing $\pi_{\omega_i}^{(m)}$. The refined in-option policy $\pi_{\omega_i}^{(m+1)}$ will be executed from state $s_{t_{m+1}}$ as the next iteration until the option is terminated or a maximum number of iterations is reached.

# 4 Experiments

## 4.1 Experimental Setup

We evaluate our framework in Minecraft, an open-ended and interactive 3D environment with a compositional action space supports diverse tasks, making it an ideal platform for assessing long-horizon reasoning and planning. Our experiment setup is built upon the Voyager (Wang et al., 2023a), which integrates its customized control primitives that can be directly called by the language model and leverages the Mineflayer JavaScript API to achieve low-level operations in Minecraft. We utilize OpenAI's GPT-4 and GPT-3.5-turbo APIs for the proposed generator and predictive modules, along with text-embedding-ada-002 for encoding options and skill descriptions for retrieval. To ensure consistency across state transition imagination, reward prediction, and feasibility assessment, all predictive modules use $temperature = 0$, a slightly higher $temperature = 0.2$ is applied during thought generation to encourage diverse reasoning.

**Long-horizon planning tasks.** Our task suite consists of eight representative objectives that capture the challenges of long-horizon reasoning in Minecraft. It covers both long-term planning and dynamic-immediate decision making, emphasizing diversity and multi-step dependencies. For example, obtaining leather requires locating animals and preparing prerequisite tools, while crafting a diamond pickaxe demands sequential resource gathering and progression through wood, stone, and iron tiers. All tasks have clearly verifiable success conditions based on the agent's inventory. Together, this suite provides a strong basis for evaluating an agent's ability to complete complex, multi-step objectives in open-ended environments.

**Baselines.** We primarily compare our framework with CoT and ToT based reasoning baselines. These methods use LLMs for planning and decomposes complex tasks into intermediate reasoning steps. Since all methods share the same underlying execution pipeline to generate code and enable interacting with the environment, our comparison focuses on the high-level reasoning capability rather than low-level perception or action control.

- **CoT** directly decomposes high-level goals into intermediate thoughts, and executes each thought through an iterative prompting loop that incorporates environment feedback, execution errors, and self-verification to refine the generated code over multiple rounds.

- **ToT-MCTS** extends the original ToT framework by generating candidate thoughts and applying basic MCTS without state transition or validation. Each node state is formalized as $s_i = [s_0, \omega_0, \omega_1, \ldots, \omega_{i-1}]$, which concatenates the initial state with the trajectory of previously generated option trace. For fairness, we provide it with the same prompting strategy for thought generation, and configure it with the same branching factor $k$ for node expansion, the same rollout depth $d$ for simulated reasoning, and the same iteration count $iter$ in MCTS.

Table 1: Performance comparison on long-horizon tasks in the Minecraft environment. "SR" refers to success rate across three execution trials, each allowing up to 30 attempts. "Time" measures the total minutes required to complete the task. "Attempts" denotes the number of low-level actor attempts (calling LLM) for code generation. Bold indicate the best results.

| Long-horizon Tasks | Model | SR | Execution Time (min) | Attempts |
|---|---|---|---|---|
| Obtain leather 🟫 | CoT | 3 / 3 | 16.17 | 14 |
| | ToT-MCTS | 2 / 3 | 12.41 | 15.5 |
| | ToO | **3 / 3** | **11.34** | **10** |
| Cook meat 🍖 | CoT | 2 / 3 | 17.65 | 23 |
| | ToT-MCTS | 3 / 3 | 14.96 | 19.67 |
| | ToO | **3 / 3** | **9.29** | **10.33** |
| Shear a sheep 🐑 | CoT | 3 / 3 | 25.10 | 21 |
| | ToT-MCTS | 2 / 3 | 17.03 | 17 |
| | ToO | **3 / 3** | **20.62** | **16.33** |
| Milk a cow 🪣 | CoT | 1 / 3 | 26.35 | 27 |
| | ToT-MCTS | 2 / 3 | 29.34 | 28 |
| | ToO | **3 / 3** | **21.8** | **21.33** |
| Collect a lava bucket 🪣 | CoT | 2 / 3 | 19.3 | 22 |
| | ToT-MCTS | 2 / 3 | 21.68 | **17** |
| | ToO | **3 / 3** | **15.7** | 17.3 |
| Craft a golden sword 🗡 | CoT | 1 / 3 | 21.95 | 28 |
| | ToT-MCTS | 1 / 3 | 29.17 | 25 |
| | ToO | **3 / 3** | **18.03** | **19.67** |
| Craft a compass 🧭 | CoT | 1 / 3 | 24.1 | 29 |
| | ToT-MCTS | 2 / 3 | 24.6 | **22.5** |
| | ToO | **3 / 3** | **16.75** | 23.3 |
| Craft a diamond pickaxe ⛏ | CoT | 0 / 3 | N/A | N/A |
| | ToT-MCTS | 1 / 3 | 25.95 | 27 |
| | ToO | **3 / 3** | **10.49** | **15.33** |

## 4.2 EVALUATION RESULTS

We evaluate our method and the baselines on a suite of long-horizon planning tasks to assess the quality of their generated plans, focusing on how effectively they decompose high-level goals into coherent and actionable reasoning trajectories.

**Long-horizon execution efficiency.** Table 1 summarizes the success rates, completion times, and the number of low-level execution attempts across eight long-horizon tasks. As shown, ToO achieves 100% success rate on all evaluated tasks, while CoT and ToT-MCTS failed on several objectives. Additionally, for the tasks they complete, ToO consistently requires the fewest execution attempts and the shortest completion time. This improvement comes from ToO's world model guided planning, where dynamics prediction and feasibility checks filter out unreliable options, resulting in more coherent plans and more efficient execution.

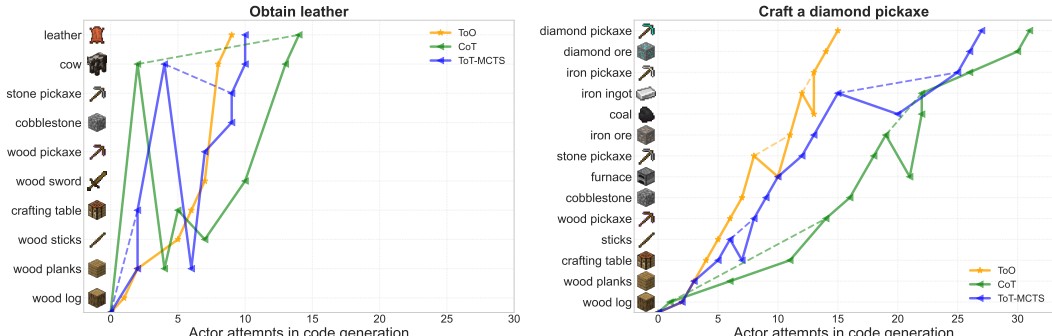

Figure 2: Comparison of execution dynamics across two categories of Minecraft long-horizon planning tasks: dynamic–immediate planning tasks (e.g., *obtain leather* 🟥) and multi-step crafting tasks (e.g., *craft a diamond pickaxe* 🛠). We visualize the execution trajectory of attempted options during code generation, where each node corresponds to one attempted option. Solid lines show the actual low-level execution sequence carried out by the actor, while dashed lines correspond to the high-level option sequence originally proposed by each method (CoT, ToT-MCTS, or ToO). The alignment or deviation between these two sequences indicates how faithfully the actor followed the planned reasoning trajectory. See Appendix A.3 for the other six tasks.

**Slow is fast.** Figure 2 provides an intuitive illustration of how different methods realize their reasoning trajectories. The x-axis measures the number of prompting attempts used for code generation during low-level execution, while the y-axis represents the sequence of options actually executed toward the objective. CoT often produces trajectories that appear faster, for example it directly proposes *find a cow* followed by *kill a cow* for the *obtain leather* task. However, without crafting the necessary tools, CoT relies on additional trial-and-error to recover the missing prerequiste steps, ultimately leading to more prompting attempts. ToT-MCTS performs a tree structured search, yet its planned trajectories resemble CoT's on long-horizon tasks, where compounding prediction errors lead to redundant actions and backtracking. In contrast, ToO combines feasibility prediction and world model rollouts during tree search. It guarantees that options are executable and incrementally progressing toward the goal, thereby avoiding invalid branches and backtracking.

**Option dependency and planning stability.** Figure 6 presents the distribution of low-level prompting attempts over the planned options, corresponding to the execution dynamics shown in Figure 2. Each bar represents the proportion of execution attempts devoted to each planned option, reflecting how heavily a method depends on specific options. CoT and ToT-MCTS exhibit relatively sparse distribution across the x-axis, but certain options dominate with disproportionately high attempt ratios. This imbalance arises from missing prerequisite steps, which causes error accumulation and results in excessively tall bars at later stages. In contrast, ToO produces a more balanced distribution of attempts over a deliberate sequence of options. By incorporating feasibility validation and dynamics prediction, ToO avoids wasting attempts on hallucinated options and instead concentrates effort on inherently high-uncertainty steps in dynamic-immediate environments, such as *find a cow*.

**Case study.** A representative long-horizon task is used to visualize the differences among the methods. As shown in Figure 3, in one trial, CoT directly proposes *find a cow* and *kill a cow*, but the missing prerequisite of *craft a wooden sword* forces repeated backtracking, resulting in 14 prompting attempts and the lowest execution efficiency. ToT-MCTS produces a more reasonable path but still overlooks essential crafting steps. In contrast, ToO leverages the predictive and simulation capabilities of the world model to imagine and infer state transitions during the search. This allows it to identify an executable option sequence that avoids backtracking and completes the task efficiently.

### 4.3 COMPUTATION AND PERFORMANCE SCALING ANALYSIS

**Pass@B evaluation.** Table 2 compares the computational costs of CoT, ToT-MCTS, and ToO. As the three methods differ greatly in its token cost $C$ per trial, SR alone cannot reflect performance

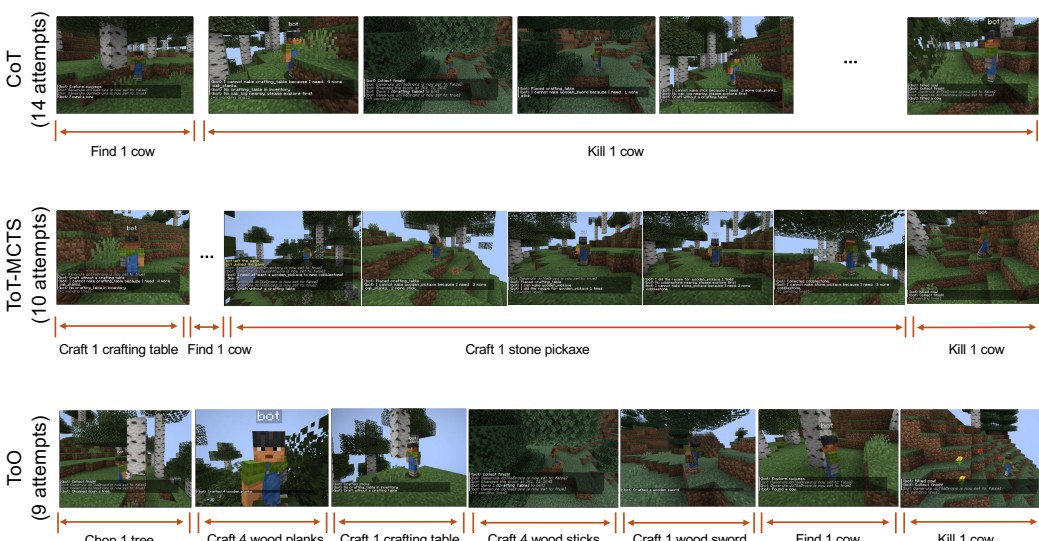

Figure 3: Comparison of how each method *obtains leather* 🟫. All three methods finish this dynamic-immediate task within 20 low-level attempts, but their reasoning traces diverge apperantly. CoT generates the shortest plan but neglects key intermediate steps. ToT-MCTS explores multiple branches and partially accounts for tool crafting. ToO leverages an LLM-based world model with feasibility checks during MCTS, yielding a backtracking-free trajectory and the most efficient execution.

under comparable computational budgets. To address this, we adopt pass@B, which provides all methods with the same total token budget $B$ and allows each method to perform $n = \lfloor B/C \rfloor$ complete trials as that budget can cover. Pass@B evaluates whether at least one trial succeeds for each task and averages this success across our task suite. Under an equal computational budget, ToO reaches a pass@B of 1.0, while CoT and ToT-MCTS complete 75% and 62.5% of tasks respectively. These results highlight ToO's superior reliability compared to the two baselines.

Table 2: Computational costs and budget-normalized success rates on long-horizon tasks. "API Calls" and "Tokens" report the number of LLM queries and the corresponding token usage during planning and execution. "Pass@B" denotes the fraction of tasks successfully solved under a shared token budget $B$, set to the cost of one ToO trial. Under this budget, CoT can run two full trials per task, while ToT-MCTS and ToO each perform one.

| Method | Planning | | Execution | | pass@B |
|---|---|---|---|---|---|
| | **API Calls** | **Tokens** | **API Calls** | **Tokens** | |
| CoT | 0.001k | 0.5k | 0.06k | 194k | 0.75 |
| ToT-MCTS | 0.6k | 215k | 0.06k | 175k | 0.625 |
| ToO | 0.9k | 347k | 0.03k | 112k | 1.0 |

**Scalability of MCTS Configurations.** Figure 4 shows how the branching factor, rollout depth, and iteration budget jointly influence MCTS behavior. The lower-left cluster of circular markers indicates the need for a larger branching factor ($k = 3$) for long-horizon tasks. A shallow depth such as $d = 8$ provides insufficient lookahead and forces the actor to compensate for missing prerequisites during execution. In contrast, an excessive depth ($d = 16$) expands the search tree too aggressively and significantly increases computational cost. A moderate depth ($d = 12$) provides the best balance, achieving high success with manageable cost. In addition, with only 5 iterations, the planner consistently fail except at $d = 12$. Increasing $iter$ to 10 yields stable success across several depths. However, even 15 iterations remain insufficient for $d = 16$, indicating that very deep trees require substantially more simulations to be effective. Overall, both ToT-MCTS and ToO benefit

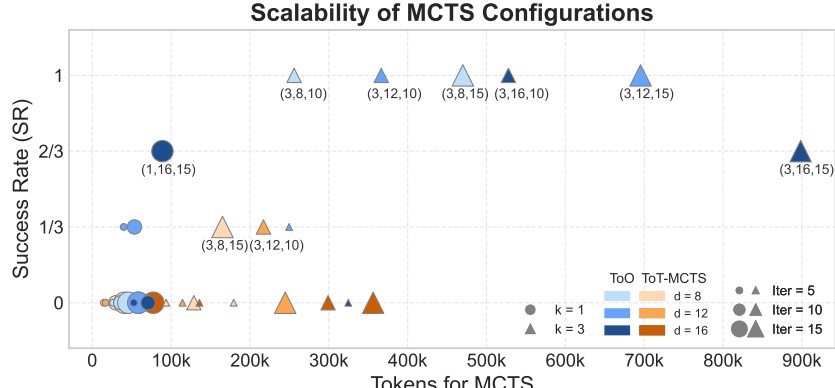

Figure 4: Scalability analysis of MCTS planners under different branching factor $k$, rollout depth $d$, and iteration count $iter$. Each point represents a configuration $(k, d, iter)$, plotted by its token consumption (x-axis) and success rate (SR) (y-axis) on the *craft a diamond pickaxe* ⛏ task. The branching factor $k$ is distinguished by marker shape (circle for $k$=1, triangle for $k$=3), rollout depth ($d = 8, 12, 16$) is indicated by increasing color intensity, and the number of iterations ($iter = 5, 10, 15$) is encoded through marker size, increasing from small to large. The results show how performance scales with computation for both ToO (blue) and ToT-MCTS (orange).

from a larger branching factor ($k = 3$), a moderate rollout depth ($d = 12$), and an adequate iteration budget ($iter = 10$), accordingly, all comparative experiments use this unified configuration.

Table 3: Ablation on ToO components.

| Long-horizon Tasks | Model | SR | Execution Time (min) | Attempts |
|---|---|---|---|---|
| Obtain leather 🛡 | ToO | **3 / 3** | **11.34** | **10** |
| | ToO w/o world model | 2 / 3 | 15.28 | 18.5 |
| | ToO w/o feasibility check | 3 / 3 | 14.97 | 13.33 |
| Cook meat 🍖 | ToO | **3 / 3** | **9.29** | **10.33** |
| | ToO w/o world model | 3 / 3 | 13.81 | 18 |
| | ToO w/o feasibility check | 3 / 3 | 13.90 | 18.33 |
| Shear a sheep 🧱 | ToO | **3 / 3** | **20.62** | **16.33** |
| | ToO w/o world model | 3 / 3 | 19.17 | 19.67 |
| | ToO w/o feasibility check | 2 / 3 | 18.73 | 17.5 |
| Milk a cow 🪣 | ToO | **3 / 3** | **21.8** | **21.33** |
| | ToO w/o world model | 2 / 3 | 27.77 | 22.5 |
| | ToO w/o feasibility check | 1 / 3 | 27.47 | 27 |

## 4.4 ABLATION STUDIES

We conduct ablation studies to evaluate the contribution of the LLM-based world model and the feasibility check module in our ToO framework.

- **ToO w/o world model** removes the LLM-based world model and uses the original ToT state $s_i = [s_0, \omega_0, \omega_1, \ldots, \omega_{i-1}]$ in tree search without state transition. Feasibility check module is still used to generate prerequisite steps during the tree search.

- **ToO w/o feasibility check** uses the same LLM-based world model to simulate transitions as ours but does not incorporate feasibility check during planning. The MCTS search expands all nodes without checking the execution constraints.

As shown in Table 3, the LLM-based world model and feasibility check module both play key roles in our framework. Specifically, removing the world model leads to relying solely on the initial input state and historical thought sequences, which results in a lack of information about environmental dynamics in the generated paths, significantly reducing planning accuracy. Even with the assistance of feasibility checks, due to single-step backtracking, it is still difficult to avoid deviations in planning. Conversely, removing the feasibility check module, while still able to imagine environmental dynamics, results in the search process extending unexecutable branches and causing redundant attempts. The combination of the two yields optimal performance and leads to higher task success rate and path efficiency. Qualitative comparisons are provided in Appendix A.3.

## 5 RELATED WORK

**Temporal abstractions.** Building agents that solve decision-making tasks in a hierarchical way is a long standing topic. The options framework proposed by Sutton et al. (1999); Precup (2000) formalizes the notion of temporally extended actions in MDPs and for RL. Prior works on options mainly focus on option discovery, where the in-option policy and/or the termination function associated with an option is not given a priori but learned through trial-and-error RL, most commonly using gradient-based approach (Sorg & Singh, 2010; Comanici & Precup, 2010; Levy & Shimkin, 2011; Silver & Ciosek, 2012; Bacon et al., 2017). With minor differences, another line of works refer to such temporally extended actions as subgoals (Kulkarni et al., 2016; Tessler et al., 2017; Vezhnevets et al., 2017), which typically adopt a manager-work architecture where the manager prescribes subgoals for the work to achieve with primitive actions. In contrast, this work implements temporally extended actions with LLMs, where options are represented as natural language (instead of abstract symbols), so that their semantics such as in-option policies and termination conditions are generated by LLMs with their commonsense knowledge and coding capabilities. In this sense, this work follows recent ones on using LLMs to plan out options/subgoals for robots and game agents (Ahn et al., 2022; Huang et al., 2022b;a; Wang et al., 2023a; Liu et al., 2024). These prior works either need RL that requires a lot of samples or employs only CoT-like planning, while we are the first to plan options/subgoals with tree search and executes them using LLMs only without extensive trial-and-error RL.

**World modeling and tree search with LLMs.** Existing work is relatively limited on repurposing LLMs as world models or heuristics policies in a way that can be incorporated into tree search algorithms like MCTS. Zhang et al. (2023) use LLMs as a heuristics policy to guide MCTS for code generation. Hao et al. (2023); Zhao et al. (2023) are the first to repurpose LLMs as a world model (i.e., transition and reward functions), which is incorporated by MCTS to solve reasoning and planning tasks. Zhou et al. (2023) propose LATS, which expands LLM-generated actions through environment interaction without relying on a world model. As the key difference, these works operate with primitive actions with no abstractions, while our ToT leverages (only) LLMs to perform world modeling and planning over the temporal abstraction of options.

**Agents for Minecraft.** Minecraft is a popular open-world sandbox game with flexible mechanics that has been serving as a benchmark for building efficient and generalized agents. We refer to Liu et al. (2024); Wang et al. (2023a) and references therein for a discussion on endeavors on Minecraft that do not leverage LLMs. This work follows recent ones that leverage LLMs as a high-level planner in Minecraft by decomposing the task into subgoals/options (Wang et al., 2023b; Nottingham et al., 2023; Yuan et al., 2023; Wang et al., 2023a). Unlike these works, we employs tree search as the planning algorithm over options that is enabled and enhanced via our option-driven world modeling, all implemented via LLMs only.

## 6 CONCLUSION

To conclude, we propose Tree of Options (ToO), a novel paradigm that leverage pretrained large language models (LLMs) to perform world modeling and tree search to plan over temporally extended actions, i.e., options, and then execute the planned out options via code generation. ToO is particularly designed to tackle complex, long-horizon embodied decision-making tasks. Our experiments demonstrate its superior performance in such tasks in the game of Minecraft.

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

# A  APPENDIX

## A.1  METHOD DETAILS

### A.1.1  STATE REPRESENTATION

We define the state $\hat{s}_i$ as a collection of attributes of the environment and the agent's state in Minecraft. It includes the inventory and equipment, the sets of nearby blocks and entities within a 32 block radius, and any recently seen blocks or visible chests, the current biome and time of day, the agent's health and hunger levels, and its position given by coordinates $(x, y, z)$.

---

**Agent state space**

Biome:
Time: night
Nearby blocks: water, sand, dirt, stone, grass_block, kelp, coal_ore, birch_log, seagrass, copper_ore, iron_ore
Nearby entities (nearest to farthest): cod
Health: 20.0/20
Hunger: 20.0/20
Position: x=198.7, y=64.0, z=116.5
Equipment: [None, None, None, None, 'crafting_table', None]
Inventory (4/36): {'stick': 2, 'wooden_pickaxe': 1, 'birch_planks': 3, 'crafting_table': 1}
Chests: None

---

### A.1.2  THOUGHT GENERATION

We formulate each option as a tuple by [verb] [quantity] [item], where the verb specifies the type of operation (e.g., mine, craft, smelt), the quantity denotes how many units involved, and the item identifies the target object. This structured action can be directly mapped to executable low-level skill library.

---

**Thought generation prompt**

You are an assistant that proposes {k} possible next subgoals that continue progress toward the final goal.

Minecraft situation:
Current State:{state}
Goal: {goal}
History actions: {action_sequence if action_sequence else "None"}

You must follow the these rules:
1. Each "task" must be a single concise actionable subgoal in the form: Mine [quantity] [block], Craft [quantity] [item], Smelt [quantity] [item], Cook [quantity] [food], Equip

---

[item].
2. Do NOT generate abstract actions such as: explore, prepare, look for, search, go to, wait.
3. Do NOT repeat steps already completed in history.
4. Do NOT include unnecessary items (weapons, armor, torches) unless required by the goal.
5. Prioritize resource gathering, tool crafting, material processing, and crafting items for the final objective.
6. Hunger is irrelevant unless directly part of the goal.
7. A subgoal is always one concrete step; you do not need to break it into pre-steps (the skill library handles that).

You should only respond in JSON format as described below:

Example format:
[
{"reasoning": "To craft a bucket, you need iron.", "task": "Mine 3 iron ore"},
{"reasoning": "To progress toward goal, ...", "task": "..."},
{"reasoning": "...", "task": "..."}
]

### A.1.3 WORLD MODEL EXPANSION

The LLM-based world model predicts the next state $\hat{s}_{i+1}$ by given the current state $\hat{s}_i$ and the proposed option $\omega_i$, while adhering strictly to Minecraft mechanics.

**Dynamics prediction prompt**

You are simulating a Minecraft world state transition. Predict exact state changes from the action.

I will give you the following information:
Current State:{state}
Proposed action: {thought}

You must follow the these rules:
1. Exploration (find, search, explore) ONLY changes nearby_entities/blocks, NEVER inventory.
2. Crafting: consumes materials, adds products to inventory.
3. Mining adds items to inventory (if proper tool available).
4. Animal interaction (milk, shear, collect) requires specific items/animals, changes inventory.
5. If requirements not met, return original state unchanged.

Output: Pure JSON only. Start with {, end with }.

### A.1.4 REWARD EVALUATION

To evaluate how each intermediate step contributes to overall task progress, we define multiple questions to assess each thought. The evaluation prompt takes the agent's current state, the sequence of previously generated thoughts, and the ultimate task goal as input. This prompt enables a comprehensive in-context assessment of whether a thought is logically coherent with the reasoning history and whether it effectively promotes the agent toward task completion.

---

**Reward prediction prompt**

You are evaluating a predicted state in a Tree-of-Thought process for Minecraft.
I will give you the following information:

Action: {thought}
Predicted State: {state}
Goal: {goal}
History actions: {action_sequence if action_sequence else "None"}

You should score the predicted state (0.0–1.0) on the following criteria:
[Success Probability]: Likelihood of eventually reaching the goal from this state.
[Reward Estimate]: Value of current progress toward the goal.
[Feasibility]: Whether the predicted state is realistic in Minecraft.
[Consistency]: Whether the action follows logically from history without skipping required steps.
[Relativity]: Relevance of the action to the goal (penalize unnecessary steps).
[Reachability]: Can this state be reached from the previous state using the action, given inventory/tool prerequisites?
[Task Specificity]: How concrete and actionable the action is (e.g., "Craft stone pickaxe" vs vague ideas like "find resources").
[Redundancy/Overkill]: Penalize redundant or unnecessarily high-tier upgrades for the goal.
[Reasoning]: Short explanation.

Strictly follow the format and provide all fields with numeric scores.

---

### A.1.5 FEASIBILITY CHECK

To further guarantee that candidate thoughts are executable within the Minecraft environment, we incorporate a feasibility check module. This component evaluates whether a proposed action or intermediate thought can be carried out given the agent's current state and available resources. By doing so, it prevents the agent from hallucinating unrealistic traces (e.g., *collect milk* by attempting *find a cow* and *milk a cow* without crafting the required bucket).

---

**Feasibility prediction prompt**

You are checking the feasibility of a Minecraft action.

I will give you the following information:
Current state:{state}
Proposed action: {thought}

You must follow the following criteria:
1. Check if required materials exist in inventory.
2. Check if required tools/entities are available.
3. Determine if action is feasible.

Example check:
Action: craft bucket.
Check: Count iron ingots in inventory.
- If the count is less than 3, it is not feasible (insufficient materials).
- If the count is at least 3, it is feasible (sufficient materials available).

Return only the feasibility assessment in this exact format:
{
"action_valid": true/false,

---

> "reason": "explain why the action is or is not feasible."
> }
>
> Return only the feasibility assessment.

When an action is deemed infeasible, we leverage the explanation generated by the feasibility prompt to guide the LLM in producing the corresponding prerequisite steps, thereby enabling the agent to recover by explicitly addressing the missing conditions.

### A.1.6  MCTS RESULT

Based on the MCTS method described in Section 3.1, we perform 10 search iterations on the shear a sheep task. According to the final tree structure, we can greedily select the nodes with the highest Q-values to derive the optimal planning path.

---
**Reasoning tree of shear a sheep**

```
ROOT (N:10, Q:4.520)
1. Punch tree to obtain 4 wood logs (N:5, Q:5.093)
1.1. Mine 8 cobblestone (N:3, Q:4.983)
1.1.1. Craft 1 stone pickaxe (N:3, Q:4.953)
1.1.1.1. Chop 1 tree to obtain wood logs (N:2, Q:4.873)
1.1.1.1.1. Craft 1 furnace (N:2, Q:4.876)
1.1.1.1.1.1. Mine 3 iron ore (N:2, Q:4.802)
1.1.1.1.1.1.1. Mine 8 coal (N:2, Q:4.657)
1.1.1.1.1.1.1.1. Smelt 3 iron ore (N:1, Q:3.955)
1.1.1.1.1.1.1.1.1. Craft shears (N:1, Q:3.164)
1.1.1.1.1.1.1.1.2. Craft 4 wooden planks (N:0, Q:0.000)
1.1.1.1.1.1.1.1.3. Find 1 sheep (N:0, Q:0.000)
1.1.1.1.1.1.1.2. Smelt 2 iron ore (N:1, Q:4.953)
1.1.1.1.1.1.1.2.1. Craft 1 shear (N:1, Q:4.608)
1.1.1.1.1.1.1.2.1.1. Find 1 sheep (N:1, Q:4.117)
1.1.1.1.1.1.1.2.1.1.1. Use shear on 1 sheep (N:1, Q:3.256)
1.1.1.1.1.1.1.2.1.1.2. Equip shear (N:0, Q:0.000)
1.1.1.1.1.1.1.2.1.1.3. Shear 1 sheep (N:0, Q:0.000)
1.1.1.1.1.1.1.2.1.2. Smelt 1 iron ore (N:0, Q:0.000)
1.1.1.1.1.1.1.2.1.3. Craft 4 wooden planks (N:0, Q:0.000)
1.1.1.1.1.1.1.2.2. Smelt 3 iron ore (N:0, Q:0.000)
1.1.1.1.1.1.1.2.3. Find 1 more sheep (N:0, Q:0.000)
1.1.1.1.1.1.1.3. Look for a sheep (N:0, Q:0.000)
1.1.1.1.1.1.2. Chop 4 wood (N:0, Q:0.000)
1.1.1.1.1.1.3. Craft 5 wooden planks (N:0, Q:0.000)
1.1.1.1.1.2. Explore area (N:0, Q:0.000)
1.1.1.1.1.3. Craft 4 wooden planks (N:0, Q:0.000)
1.1.1.1.2. Mine 3 iron ore (N:0, Q:0.000)
1.1.1.1.3. Find iron ore (N:0, Q:0.000)
1.1.1.2. Craft 1 furnace (N:1, Q:4.766)
1.1.1.2.1. Explore area to find 1 sheep (N:1, Q:4.662)
1.1.1.2.1.1. Craft 1 shears (N:1, Q:4.069)
1.1.1.2.1.1.1. Use shears on sheep (N:1, Q:3.205)
1.1.1.2.1.1.2. Shear 1 sheep (N:0, Q:0.000)
1.1.1.2.1.1.3. Equip shears (N:0, Q:0.000)
1.1.1.2.1.2. Explore area (N:0, Q:0.000)
1.1.1.2.1.3. Mine 3 iron ore (N:0, Q:0.000)
1.1.1.2.2. Explore (N:0, Q:0.000)
1.1.1.2.3. Mine 3 iron ore (N:0, Q:0.000)
1.1.1.3. Find iron ore (N:0, Q:0.000)
```
---

```
1.1.2. Craft 4 wooden planks (N:0, Q:0.000)
1.1.3. Craft 16 wooden planks (N:0, Q:0.000)
1.2. Craft 4 wooden planks (N:1, Q:4.796)
1.2.1. Mine 3 iron ore (N:1, Q:4.767)
1.2.1.1. Smelt 3 iron ore (N:1, Q:4.715)
1.2.1.1.1. Craft 1 shears (N:1, Q:4.282)
1.2.1.1.1.1. Find sheep (N:1, Q:3.338)
1.2.1.1.1.2. Chop 2 wood logs (N:0, Q:0.000)
1.2.1.1.1.3. Explore landscape for a sheep (N:0, Q:0.000)
1.2.1.1.2. Craft 1 stone pickaxe (N:0, Q:0.000)
1.2.1.1.3. Craft 1 crafting table (N:0, Q:0.000)
1.2.1.2. Chop 4 wood (N:0, Q:0.000)
1.2.1.3. Craft 1 Crafting Table (N:0, Q:0.000)
1.2.2. Mine 8 cobblestone (N:0, Q:0.000)
1.2.3. Craft 1 wooden pickaxe (N:0, Q:0.000)
1.3. Craft 1 wooden pickaxe (N:1, Q:4.782)
1.3.1. Craft 4 wooden planks (N:1, Q:4.673)
1.3.1.1. Craft 4 sticks (N:1, Q:4.662)
1.3.1.1.1. Explore plains biome for sheep (N:1, Q:4.612)
1.3.1.1.1.1. Craft 1 shears (N:1, Q:4.144)
1.3.1.1.1.1.1. Equip shears (N:1, Q:3.077)
1.3.1.1.1.1.2. Mine 8 cobblestone (N:0, Q:0.000)
1.3.1.1.1.1.3. Shear 1 sheep (N:0, Q:0.000)
1.3.1.1.1.2. Mine 3 iron ore (N:0, Q:0.000)
1.3.1.1.1.3. Mine 8 cobblestone (N:0, Q:0.000)
1.3.1.1.2. Mine 8 cobblestone (N:0, Q:0.000)
1.3.1.1.3. Mine 3 iron ore (N:0, Q:0.000)
1.3.1.2. Chop 3 wood logs (N:0, Q:0.000)
1.3.1.3. Mine 8 cobblestone (N:0, Q:0.000)
1.3.2. Find gravel (N:0, Q:0.000)
1.3.3. Chop trees to obtain 2 wood logs (N:0, Q:0.000)
2. Chop 4 wood (N:5, Q:5.064)
2.1. Craft 1 wooden pickaxe (N:3, Q:5.102)
2.1.1. Mine 3 iron ore (N:3, Q:5.111)
2.1.1.1. Mine 8 cobblestone (N:3, Q:5.094)
2.1.1.1.1. Craft 1 furnace (N:2, Q:5.063)
2.1.1.1.1.1. Mine 8 coal (N:1, Q:4.865)
2.1.1.1.1.1.1. Explore grass patches for sheep (N:1, Q:4.642)
2.1.1.1.1.1.1.1. Smelt 2 iron ore (N:1, Q:4.167)
2.1.1.1.1.1.1.1.1. Craft 1 iron shears (N:1, Q:3.210)
2.1.1.1.1.1.1.1.2. Craft 4 wooden planks (N:0, Q:0.000)
2.1.1.1.1.1.1.1.3. Chop 7 wood (N:0, Q:0.000)
2.1.1.1.1.1.1.2. Craft 4 sticks (N:0, Q:0.000)
2.1.1.1.1.1.1.3. Smelt 3 iron ore (N:0, Q:0.000)
2.1.1.1.1.1.2. Smelt 2 iron ore (N:0, Q:0.000)
2.1.1.1.1.1.3. Smelt 3 iron ore (N:0, Q:0.000)
2.1.1.1.1.2. Smelt 3 iron ore (N:1, Q:4.892)
2.1.1.1.1.2.1. Craft 1 shears from iron ingot (N:1, Q:4.783)
2.1.1.1.1.2.1.1. Chop 8 wood (N:1, Q:4.177)
2.1.1.1.1.2.1.1.1. Craft 1 stone pickaxe (N:1, Q:3.843)
2.1.1.1.1.2.1.1.1.1. Find sheep (N:1, Q:2.836)
2.1.1.1.1.2.1.1.1.2. Explore around for sheep (N:0, Q:0.000)
2.1.1.1.1.2.1.1.1.3. Find 2 sheep (N:0, Q:0.000)
2.1.1.1.1.2.1.1.2. Chop 5 wood (N:0, Q:0.000)
2.1.1.1.1.2.1.1.3. Find 1 sheep (N:0, Q:0.000)
2.1.1.1.1.2.1.2. Explore the grass biome (N:0, Q:0.000)
2.1.1.1.1.2.1.3. Chop 3 wood (N:0, Q:0.000)
```

```
2.1.1.1.1.2.2. Craft 1 more shears from iron ingot(N:0,Q:0.000)
2.1.1.1.1.2.3. Explore to locate a sheep (N:0, Q:0.000)
2.1.1.1.1.3. Chop 7 wood (N:0, Q:0.000)
2.1.1.1.2. Mine 8 coal (N:1, Q:4.884)
2.1.1.1.2.1. Smelt 2 iron ore (N:1, Q:4.795)
2.1.1.1.2.1.1. Craft 4 stick (N:1, Q:4.592)
2.1.1.1.2.1.1.1. Smelt 1 iron ore (N:1, Q:4.430)
2.1.1.1.2.1.1.1.1. Find sheep (N:1, Q:3.984)
2.1.1.1.2.1.1.1.1.1. Craft 1 shears (N:1, Q:2.933)
2.1.1.1.2.1.1.1.1.2. Craft 1 stone pickaxe (N:0, Q:0.000)
2.1.1.1.2.1.1.1.1.3. Craft 4 wooden planks (N:0, Q:0.000)
2.1.1.1.2.1.1.1.2. Explore new area for sheep (N:0, Q:0.000)
2.1.1.1.2.1.1.1.3. Craft 1 shears (N:0, Q:0.000)
2.1.1.1.2.1.1.2. Craft 4 wooden planks (N:0, Q:0.000)
2.1.1.1.2.1.1.3. Craft 1 stone pickaxe (N:0, Q:0.000)
2.1.1.1.2.1.2. Craft 1 furnace (N:0, Q:0.000)
2.1.1.1.2.1.3. Chop 2 trees (N:0, Q:0.000)
2.1.1.1.2.2. Chop 3 wood (N:0, Q:0.000)
2.1.1.1.2.3. Find cobblestone (N:0, Q:0.000)
2.1.1.1.3. Chop 2 wood (N:0, Q:0.000)
2.1.1.2. Mine 8 coal (N:0, Q:0.000)
2.1.1.3. Chop 2 wood (N:0, Q:0.000)
2.1.2. Mine 8 cobblestone (N:0, Q:0.000)
2.1.3. Explore (N:0, Q:0.000)
2.2. Craft 4 wooden planks (N:2, Q:4.806)
2.2.1. Mine 8 cobblestone (N:2, Q:4.791)
2.2.1.1. Craft 1 furnace (N:1, Q:4.775)
2.2.1.1.1. Craft 1 stone pickaxe (N:1, Q:4.629)
2.2.1.1.1.1. Explore (N:1, Q:4.349)
2.2.1.1.1.1.1. Mine 3 iron ore (N:1, Q:4.207)
2.2.1.1.1.1.1.1. Smelt 3 iron ore (N:1, Q:3.856)
2.2.1.1.1.1.1.1.1. Craft 1 shears (N:1, Q:2.682)
2.2.1.1.1.1.1.1.2. Chop 2 wood (N:0, Q:0.000)
2.2.1.1.1.1.1.1.3. Chop 3 wood (N:0, Q:0.000)
2.2.1.1.1.1.1.2. Find 1 sheep (N:0, Q:0.000)
2.2.1.1.1.1.2. Chop 2 trees (N:0, Q:0.000)
2.2.1.1.1.1.3. Chop 2 wood (N:0, Q:0.000)
2.2.1.1.1.2. Chop 1 tree (N:0, Q:0.000)
2.2.1.1.1.3. Find a cave (N:0, Q:0.000)
2.2.1.1.2. Mine 3 iron ore (N:0, Q:0.000)
2.2.1.1.3. Place 1 furnace (N:0, Q:0.000)
2.2.1.2. Craft 1 stone pickaxe (N:1, Q:4.446)
2.2.1.2.1. Explore area (N:1, Q:4.286)
2.2.1.2.1.1. Mine 2 iron ore (N:1, Q:3.908)
2.2.1.2.1.1.1. Craft 1 shears (N:1, Q:3.205)
2.2.1.2.1.1.2. Mine 1 cobblestone (N:0, Q:0.000)
2.2.1.2.1.1.3. Smelt 2 iron ore (N:0, Q:0.000)
2.2.1.2.1.2. Mine 3 iron ore (N:0, Q:0.000)
2.2.1.2.2. Mine 1 cobblestone (N:0, Q:0.000)
2.2.1.2.3. Mine 3 iron ore (N:0, Q:0.000)
2.2.1.3. Mine 3 iron ore (N:0, Q:0.000)
2.2.2. Craft 1 wooden pickaxe (N:0, Q:0.000)
2.2.3. Craft 4 sticks (N:0, Q:0.000)
2.3. Mine 8 cobblestone (N:0, Q:0.000)
3. Craft 1 wooden pickaxe (N:0, Q:0.000)
```

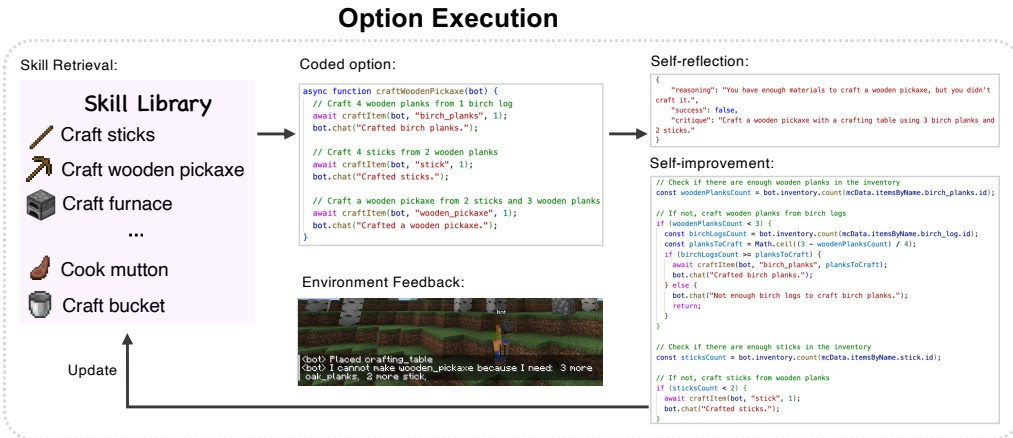

Figure 5: Option execution based on Voyager skill library.

## A.2 VOYAGER SKILL LIBRARY INTEGRATION

In order to explain how our method executes each subgoal using Voyager skill library. We use the example of *Craft a stone pickaxe* to detail the process. After connecting to Voyager inference interface, the Voyager skill manager retrieves the five most relevant skills from the learned skill library using semantic similarity matching based on text embeddings.

---

**Skill Manager retrieved skills**

craftStonePickaxe, craftIronPickaxe, craftWoodenPickaxe, craftStoneShovel, craftIronAxe

---

Then, Voyager prompts GPT-4 with contexts including the agent's current observation, the question and answer about how to achieve the subgoal, execution feedback and generated code from last round, and the retrieved skills are included as in-context references to guide code generation.

---

**Voyager prompt**

Health: 20.0/20
Hunger: 20.0/20
Position: x=209.5, y=82.0, z=141.5
Inventory: 'stick': 2, 'wooden_pickaxe': 1, 'birch_planks': 3, 'crafting_table': 1
Nearby blocks: birch_log, grass_block, stone

Task: Craft 1 stone pickaxe
Question: How to craft 1 stone pickaxe in Minecraft?
Answer: To craft 1 stone pickaxe in Minecraft, you will need 3 pieces of stone and 2 sticks. Place the 3 pieces of stone across the top row of the crafting table, and then place 2 sticks in a vertical line beneath the stones. This will create a stone pickaxe for you to use in the game.

Execution error: No error
Critique: None

Code from the last round: No code in the first round

---

Given this information, the LLM-based Voyager action agent first analyzes the task prerequisites to provide an explanation and plan, and then generates the following executable code:

> **Voyager action agent output**
>
> Explain: There is no error in the last round. The bot has 2 sticks, 1 wooden pickaxe, 3 birch planks, and 1 crafting table in its inventory. However, it doesn't have any stone, which is required to craft a stone pickaxe.
>
> Plan:
> 1) The bot will mine 3 stone blocks using the wooden pickaxe.
> 2) The bot will place the crafting table near its current position.
> 3) The bot will craft a stone pickaxe using the crafting table.
>
> ```
> async function craftStonePickaxe(bot) {
> await mineBlock(bot, "stone", 3);
> bot.chat("Collected stone.");
> const craftingTablePosition = bot.entity.position.offset(1, 0, 0);
> await placeItem(bot, "crafting_table", craftingTablePosition);
> await craftItem(bot, "stone_pickaxe", 1);
> bot.chat("Crafted a stone pickaxe.");
> }
> ```

The generated code is executed via Voyager's control primitive APIs, which are specifically designed to be interpretable and reusable, and can be directly called by GPT-4. For example, `mineBlock(bot, "stone", 3)` instructs the agent to mine three stone blocks within range, while `craftItem(bot, "stone_pickaxe", 1)` triggers the crafting of a stone pickaxe using a nearby crafting table. Internally, the lower-level primitive Mineflayer APIs, which provide direct motor control over the Minecraft agent such as `await bot.pathfinder.goto(goal)` for navigating to a specific position and `bot.useOn(entity)` for using tools on entities.

After executing the generated code, the critic agent returns its assessment based on the execution log records combined with the state, and then outputs:

> **Voyager critic agent output**
>
> ```
> {
> "reasoning": "You have a stone pickaxe in your inventory, which means you successfully crafted it.",
> "success": true
> }
> ```

## A.3 RESULTS

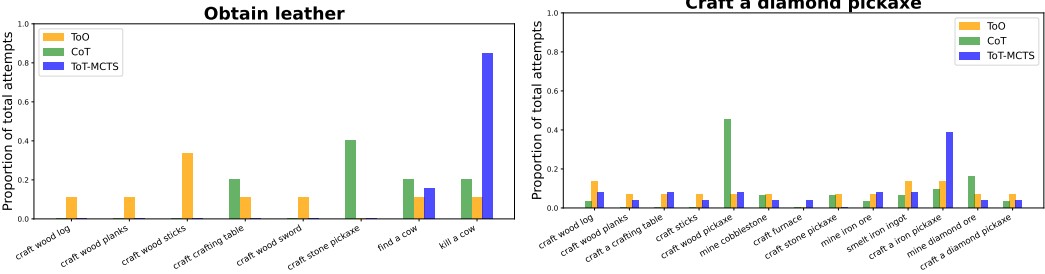

Figure 6: Comparison of execution attempts across planned options in dynamic–immediate planning task (*obtain leather* 🟫) and multi-step crafting task (*craft a diamond pickaxe* ⛏). Fewer attempts indicate the necessary prerequisites were met, while a larger proportion suggests inefficient proposals.

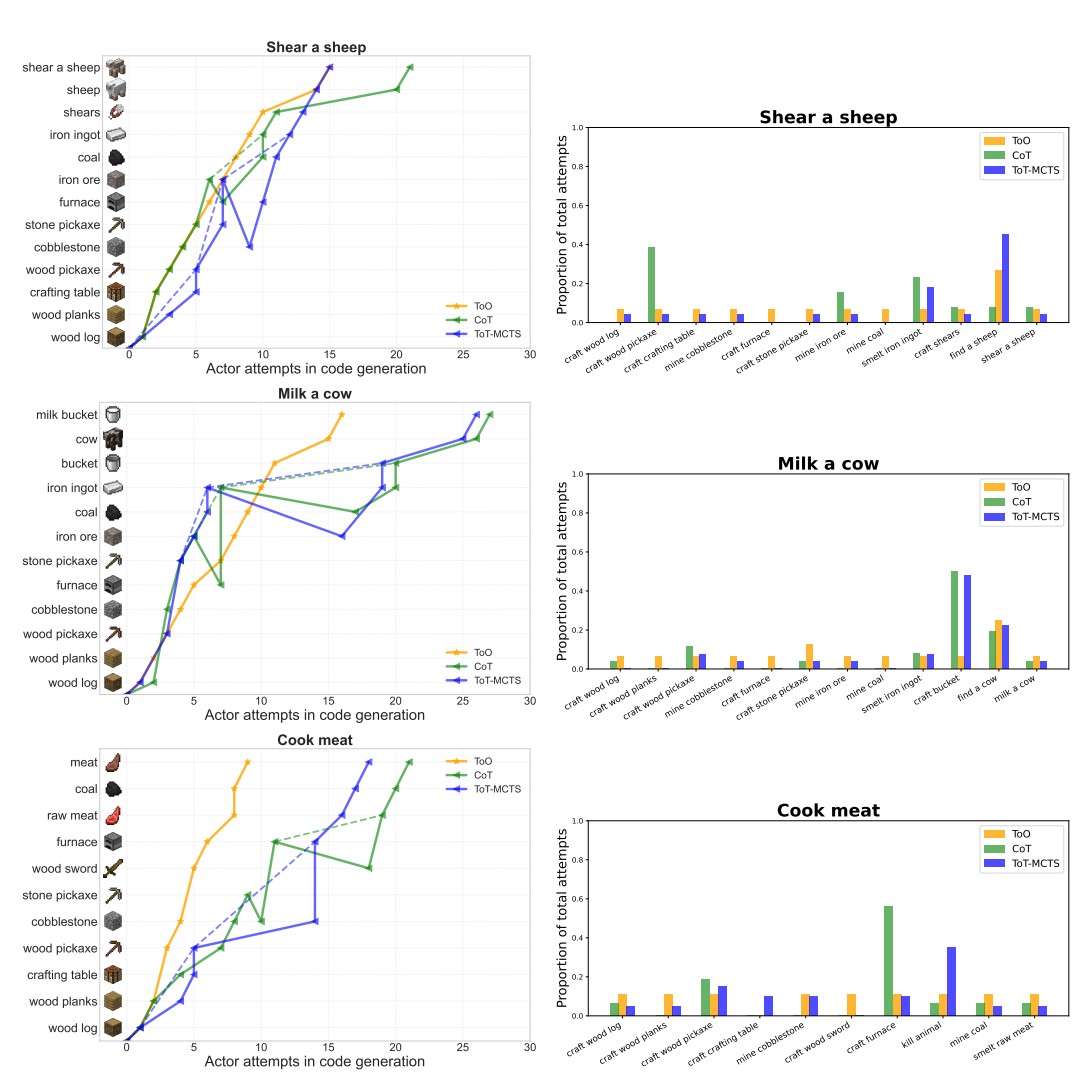

Figure 7: Comparison of execution dynamics across dynamic–immediate planning tasks.

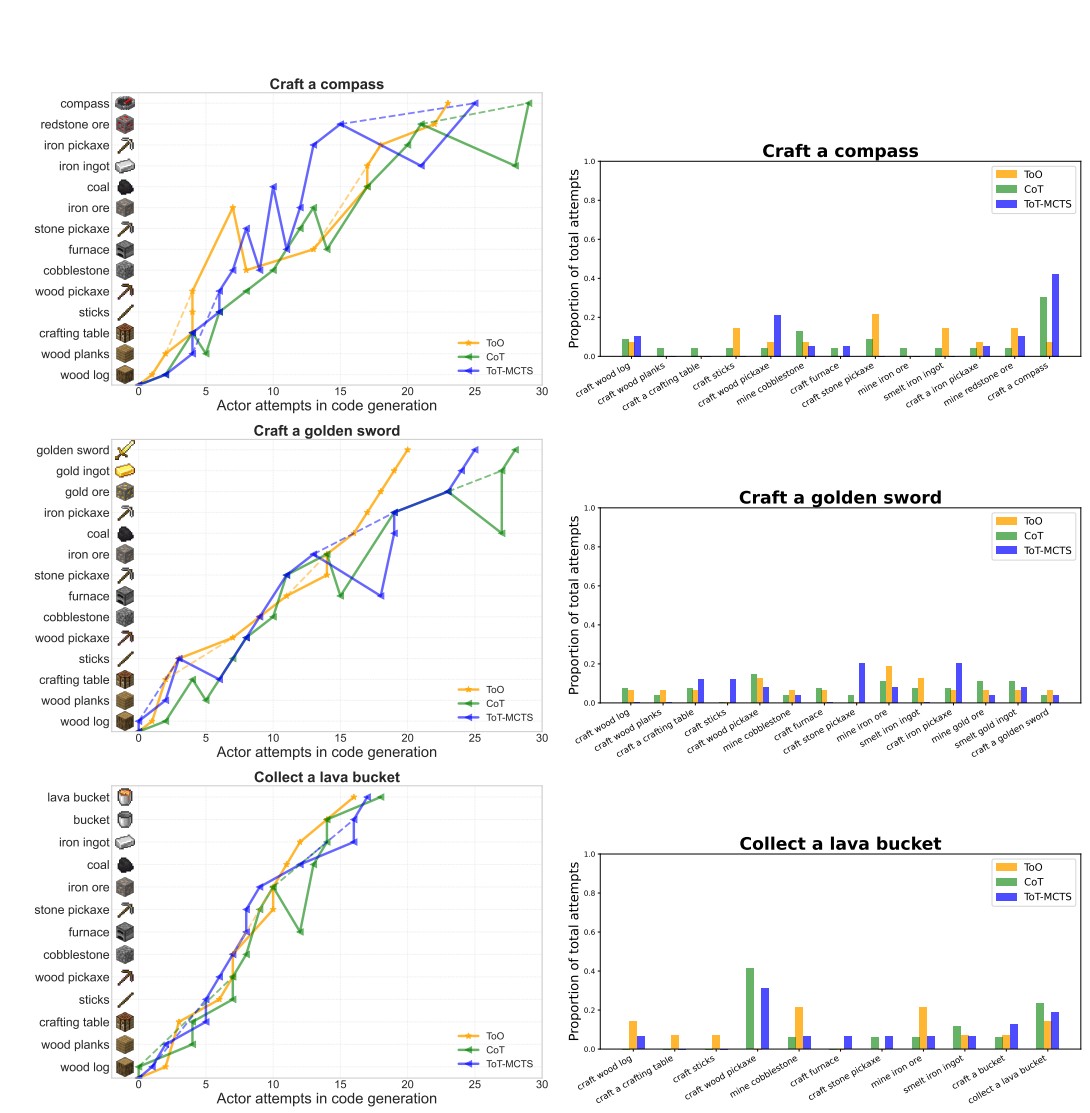

Figure 8: Comparison of execution dynamics across multi-step crafting tasks.

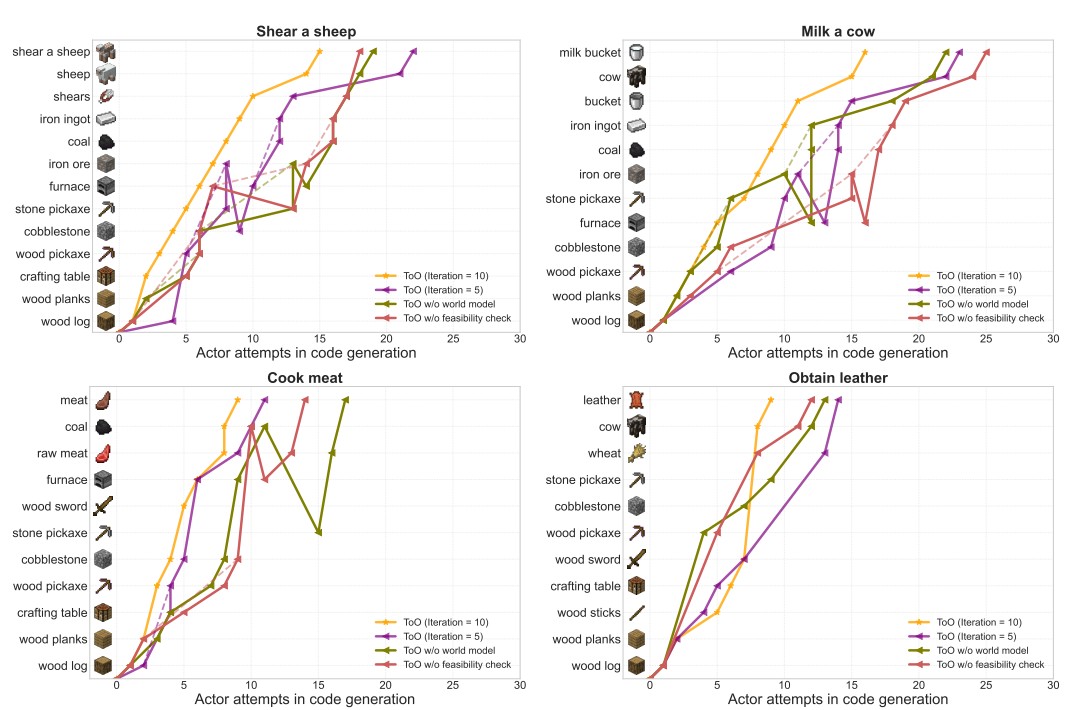

Figure 9: Ablation studies for LLM world model and feasibility check module.

