# OpenReview forum: "Tree-of-Options: Temporally Extended World Modeling, Planning, and Execution with Large Language Models"
_ICLR.cc/2026/Conference — Submitted to ICLR 2026_

### Official Review · Reviewer_t5aT · 2025-10-26

**Soundness:** 3
**Presentation:** 3
**Contribution:** 2
**Rating:** 4
**Confidence:** 3

**Summary:**

The authors propose Tree of Options (ToO), using temporally extended actions instead of primitive actions to improve large language models' performance in long-horizon, complex tasks. The authors choose Minecraft as the target environment and experiment ToO on four long-horizontal tasks against CoT and ToT+MCTS, showing promising results.

**Strengths:**

1. For long-horizon tasks, using options as instead of actions is straightforward and convincing. Options can be viewed as a higher level action comparing to primitive actions. Building the tree of options shortens the length of trajectory and facilitates LLMs to make decisions on a higher level.
2. Presentation of Figure 2 is straightforward, where ToO obtains a "flatter" distribution of lower-level attempts. Flatter distribution of lower-level attempts could potentially help LLMs jump out of local maxima and go to the ultimate goal faster.

**Weaknesses:**

1. Contribution is limited. I would like to view ToO as a variant of Language Agent Tree Search (LATS) [1], where the actions are replaced by options, where LLM initially critics on options and then try implements with actual feedbacks with environments.

2. The experiment results partially supports the argument, but not fully. The long-horizon tasks should be harder and longer, like some tasks where the Diamond tools are involved. Also, for all the tasks, the baselines (CoT and ToT + MCTS) are able to achieve successfully. Some results that the baselines cannot achieve within some reasonable extended budgets comparing to ToO would support the authors' argument much better, where options do help LLMs perform reliably in long-term, complex tasks.

3. The implementation of options is also limited. The authors use a prebuilt Voyager [2] skill library to implement Minecraft, and a trial-error fashion style based on the semantic similarity between options and skill library function names. While those implementations are simple and straightforward, a more general or principled way to implement execution of options would be appreciated.

4. The related work section needs to be better. For instance, this paper is not cited: "Mastering Board Games by External and
Internal Planning with Language Models" [3].

[1] Zhou, Andy, et al. "Language agent tree search unifies reasoning acting and planning in language models." arXiv preprint arXiv:2310.04406 (2023).
[2] Wang, Guanzhi, et al. "Voyager: An open-ended embodied agent with large language models." arXiv preprint arXiv:2305.16291 (2023).
[3] Schultz, John, et al. "Mastering board games by external and internal planning with language models." arXiv preprint arXiv:2412.12119 (2024).

**Questions:**

1. Would you mind providing additional experiments results on longer horizontal tasks? (Connection to weakness 2)

2. For the option, how many actor code generation attempts needed to fully implement the option?

3. What's the difference between option and action? For me, it seems like the options would be slightly general queries to retrieve from Voyager [1] skill library for code implementations comparing to detailed verbal actions. This is a nice contribution with simple solutions but there's no fundamental differences between options and actions.

4. What's the cost comparison among CoT, ToT + MCTS, ToO (iteration =5) and ToO (iteration = 10)? Dollar computation would be sufficient (including OpenAI embedding model costs for ToO).

[1] Wang, Guanzhi, et al. "Voyager: An open-ended embodied agent with large language models." arXiv preprint arXiv:2305.16291 (2023).

---

> ### Author Response · Authors · 2025-12-03
>
> Thank you for reviewing our paper!
> Please find below our response to the Weaknesses (W) and Questions (Q).
>
> **(Q3) Option vs action**
> As our first response, we clarify the difference of option vs action:
> - Actions are defined in the original agent-environment RL problem: they are primitive decisions/controls to the transition dynamics, operating on every “tick” of the environment.
> - Options are an algorithmic framework to provide temporal abstraction: an option specifies i) initial and terminal conditions to (de)activate it and ii) an in-option policy, i.e., how to select actions in each state when the option is active. Under this framework, an agent executes the current option by following its in-option policy until termination, at which point the agent picks the next option to execute. Therefore, usually options are conceptually equal to skills/subgoals.
> - For Minecraft, actions are mouse and keyboard inputs (e.g., turn left, pick-up, move forward). Voyager defined a particular skill/option library, but other libraries are also valid for Minecraft.
> - Our main contribution lies in the effective planning of options (Sec. 3.1). The experiments took advantage of Voyager’s library for option implementation/execution, yet our ToO framework is compatible with any option implementation/execution method.
>
> **(W3) Implementation of options**
> The method described in Sec. 3.2 is general and principled for any LLM-based option implementation, including Voyager as a special case as it uses LLMs for its library.
>
> Voyager is the state-of-the-art for LLM-based Minecraft agents. Using it in our experiments ensures fair comparison and isolates the effectiveness of our ToO’s option-level world modeling and planning.
>
> Because our ToO framework separates options planning from implementation, it is compatible with other, non-LLM implementations of options, and we agree they could further demonstrate the usefulness of ToO.
>
> **(Q2) Actor code generation attempts needed**
> In our experiments, we put a limit on the total number of attempts to execute the full sequence of planned options (which is 30); we did not limit the number of attempts allocated for a single option.
>
>
> **(W1) Comparison with works like LATS**
> Thank you for suggesting LATS.
> LATS, like RAP [Hao et al., 2023] that we cited, performs world modeling (transition and reward) with LLMs and planning with it.
> Yet, prior works like LATS and RAP do not involve options, or any form of temporal abstraction, which is the key difference of our work.
>
> **(W4) Related work, Schultz et al. 2024**
> Thank you for suggesting [Schultz et al. 2024] on LLMs for board games. We have included it in our revision.
> In short, for board games like Chess, it is non-trivial, if not impossible, to design/discover useful temporal abstractions like options. Therefore, there is little existing work in this line: [Schultz et al. 2024] does not involve options and our work does not involve board games.
>
> Our related work section has discussed works most related to ours, and we will incorporate other related works if suggested.
>
> **(W2, Q1) Results on longer, harder tasks**
> Our revision provides further results on longer, harder tasks, including a Diamond task. Please refer to our official comment titled “Summary of Revisions" for details.
>
>
> **(Q4) Cost analysis**
> Our revision provides further results on cost analysis. Please refer to our official comment titled “Summary of Revisions" for details.

---

### Official Review · Reviewer_HKvo · 2025-10-29

**Soundness:** 2
**Presentation:** 1
**Contribution:** 2
**Rating:** 2
**Confidence:** 3

**Summary:**

This paper introduces the Tree of Options (ToO) framework, a novel planning method designed to enhance the performance of Large Language Models (LLMs) on long-horizon embodied decision-making tasks. The framework integrates Monte Carlo Tree Search (MCTS) with a high-level, LLM-based world model to plan over an abstract space of temporally extended actions, or "options." Experiments conducted in the Minecraft environment demonstrate that ToO achieves superior efficiency and reliability on complex tasks when compared to baseline methods.

**Strengths:**

- By planning at the "option" level, the framework astutely leverages the high-level reasoning strengths of LLMs while mitigating their known weaknesses in handling low-level control details. The integration of MCTS with an option-level world model is a well-motivated and insightful contribution.
- The choice of Minecraft as a testbed for long-horizon tasks is highly appropriate. Furthermore, the comparison against the ToT-MCTS baseline effectively ablates and highlights the core contribution of the option-level world model.

**Weaknesses:**

- The framework involves multiple nested LLM calls within a single MCTS iteration, which appears computationally expensive. The paper does not discuss the associated costs or the scalability of this approach.
- The success of each component (e.g., dynamics model, feasibility check) is highly dependent on carefully engineered prompts. This approach can be brittle and may require significant re-engineering for new domains or even for slightly different tasks, raising questions about its generalizability.
- While "actor attempts in code generation" is a useful proxy for efficiency, the evaluation would be more persuasive if it also prominently featured standard metrics such as overall task success rate, total wall-clock time, and the total number of LLM API calls, some of which are currently in the appendix.

**Questions:**

- Given that ToO involves multiple nested LLM calls per MCTS iteration, could the authors provide a quantitative analysis of the planning costs (e.g., number of API calls, token consumption, latency) and discuss the framework's scalability as the number of MCTS iterations and the branching factor increase?
- The current implementation relies on one-shot planning. How might the system adapt to significant discrepancies between the world model's prediction and the actual outcome of an action? A discussion of strategies for online replanning and the associated trade-offs with the high planning cost would be valuable for assessing the framework's practicality.
- Could the evaluation be made more comprehensive by supplementing the primary metric ("number of code generation attempts") with standard metrics like task success rate and total execution time in the main text?
- It would be interesting to hear the authors' perspective on the trade-offs of the iterative code generation approach—a form of trial-and-error learning—compared to traditional reinforcement learning methods for executing complex options.

---

> ### Author Response · Authors · 2025-12-03
>
> Thank you for reviewing our paper!
> Please find below our response to the Weaknesses and Questions.
>
> **Cost and scalability analysis**
> Our revision provides further results on cost and scalability analysis. Please refer to our official comment titled “Summary of Revisions".
>
> **Standard performance metrics**
> As suggested, our revision includes summary tables of standard metrics in the main body.
> Please refer to our official comment titled “Summary of Revisions" for details.
>
> **Prompt engineering, generalizability**
> We hand-engineered the prompts yet did not perform extensive prompt engineering, except for the reward prediction module where we hand-engineered the evaluation questions and weights for Eq. (6) with a few trial-and-error iterations.
> For all the other modules, the prompts were designed following the same practice as prior work such as Voyager and RAP.
> Further, for fair comparison, the baselines (CoT, ToT-MCTS) use the same prompts whenever possible.
>
> We agree that reducing prompt dependence or learning prompts automatically would further strengthen our method, yet we view such efforts to be general to any LLM-based methods and not specific to our ToO framework.
>
> **On online replanning**
> We agree that a study on online replanning would be valuable, as it might reduce the planning cost.
> Our ToO framework is compatible with online replanning in a straightforward manner, yet the best efficiency relies on ideal timings of replanning, which might be highly task-specific and need additional hyperparameter tuning.
>
>
> **Iterative code generation vs traditional reinforcement learning**
> We thank the reviewer for raising this issue.
> It is a general question. We provide a few general points from our perspective and link to our work whenever appropriate:
> - Traditional reinforcement learning (RL) offers rigorous formalisms and methods. The hope is that, with sufficiently large sample and computation budgets, agents will be able to identify the optimal decisions, typically by learning from scratch. The drawbacks are well known: traditional RL is sample- and compute-inefficient, sensitive to hyperparameters, and often struggles for sparse-reward and long-horizon tasks.
> - Iteratively generating code as policies by LLMs offers a different alternative. It avoids gradient-based RL entirely. The policy represented by code is more interpretable. Most importantly, by leveraging LLMs’ pretrained capabilities, it is much more efficient and effective for “common-sense” tasks. Yet, it is difficult to provide formal guarantees (e.g., convergence, optimality).
> - Therefore, the two alternatives are complementary. Traditional RL should be preferred when one is interested in finding (near-)optimal solutions or the task of interest exhibits little common sense that can be exploited by LLMs; while code generation should be preferred when “good enough” solutions are acceptable for a common-sense task.
> - Clearly, for Minecraft, iterative code generation is advantageous. We are not super interested in reducing the execution time from, say, 7min to 6min59s. LLMs are well capable of generating code that perform skills and small  in Minecraft
> - Combining the best of the two worlds would be an exciting future direction.

---

### Official Review · Reviewer_yJ7s · 2025-10-30

**Soundness:** 3
**Presentation:** 2
**Contribution:** 2
**Rating:** 4
**Confidence:** 2

**Summary:**

This paper introduces Tree of Options (ToO) — a framework that integrates Large Language Models (LLMs) into temporally abstract world modeling and planning via Monte Carlo Tree Search (MCTS). Unlike previous LLM-based planners that operate on primitive actions (leading to compounding prediction errors), ToO models temporally extended actions (“options”) in natural language form.

**Strengths:**

1. Introduces a language-level option framework—bridging classical hierarchical RL (options) and LLM reasoning.
2. Integrates LLMs into MCTS for structured exploration and evaluation over temporally extended actions. The combination of option-driven dynamics and reward predictors gives a coherent planning architecture.
3. Uses LLMs’ strength in program synthesis to translate abstract options into executable skills. Iterative refinement with feedback increases robustness against generation errors.

**Weaknesses:**

1. The approach depends on carefully hand-engineered prompts (for option generation, feasibility checks, reward prediction, etc.).
2. While qualitative trajectories are shown, there is limited quantitative analysis on computational cost, token usage, or LLM query efficiency compared to baselines.
3. The paper lacks a formal discussion of convergence, optimality guarantees, or the relationship between option-level abstraction depth and search efficiency.
4. Ablations isolate world modeling and feasibility checks but do not examine how MCTS hyperparameters, rollout length, or option vocabulary size affect performance

**Questions:**

More discussions about Weaknesses.

---

> ### Author Response · Authors · 2025-12-03
>
> Thank you for reviewing our paper!
> Please find below our response to the Weaknesses (W).
>
>
> **(W1) Prompt engineering (option generation, feasibility checks, reward prediction, etc.)**
> We hand-engineered the prompts yet did not perform extensive prompt engineering, except for the reward prediction module where we hand-engineered the evaluation questions and weights for Eq. (6) with a few trial-and-error iterations.
> For all the other modules, the prompts were designed following the same practice as prior work such as Voyager and RAP.
> Further, for fair comparison, the baselines (CoT, ToT-MCTS) use the same prompts whenever possible.
>
> We agree that reducing prompt dependence or learning prompts automatically would further strengthen our method, yet we view such efforts to be general to any LLM-based methods and not specific to our ToO framework.
>
>
> **(W2) Quantitative analysis**
> Thank you for raising this point. We agree quantitative analysis would enhance this work.
> Our revision has included comprehensive quantitative results. Please refer to our official comment titled “Summary of Revisions" for details.
>
> **(W3) Convergence/optimality guarantees, search efficiency**
> Thank you for the suggestion.
> It would be highly-nontrial to provide formal convergence/optimality guarantees for LLM-based planning and execution, as we are not aware of such guarantees in prior works either.
>
> Our revision provides further results on search efficiency and scalability analysis. Please refer to our official comment titled “Summary of Revisions".
>
> **(W4) Effect of hyperparameters**
> Thank you for raising this point.
> Our revision includes results on the effect of MCTS hyperparameters, for both ToT-MCTS and our ToO as a comparison.
> In short, our ToO scales much better with the search hyperparameters. Please refer to our official comment titled “Summary of Revisions".
>
> We would like to note that option vocabulary size is not a hyperparameter, as the options are represented as natural language and directly planned and executed by LLMs, which is a crucial difference between our work and traditional option frameworks.

---

### Official Review · Reviewer_VxES · 2025-11-01

**Soundness:** 2
**Presentation:** 1
**Contribution:** 2
**Rating:** 2
**Confidence:** 4

**Summary:**

This paper presents ToO, a framework that lets LLMs plan and act through temporally extended actions instead of step-by-step moves. It combines an LLM-based world model, which predicts how the environment changes when a high-level option is executed, with a MCTS  planner that selects the best option sequence. The system then prompts the LLM again to generate executable code for each option. Experiments in Minecraft show that it handles long, multi-step tasks more reliably than Chain-of-Thought or Tree-of-Thought baselines, producing steadier and more feasible plans.

**Strengths:**

- The pape introduces the options concept from RL into the context of LLM-based planning.

- The experiments provid qualitative analyses of the slow is fast, option dependency stability, and the role of feasibility validation, which make the behavioral insights richer.

**Weaknesses:**

- The overall presentation of the paper could be improved. Figure 1 lacks clear annotations and does not clearly illustrate how the world modeling component is integrated into the framework.

- The main results section relies heavily on visualizations (Figures 2–4). As these figures are not sufficiently explained, it takes some effort for me to understand their logic. Since they mainly show a few specific tasks, the results feel more like case studies. It would greatly strengthen the paper to include a summary table with clear metrics such as task success rate or average step length.

- All experiments are conducted in the Minecraft environment with very similar task settings (mining, crafting, milking, etc.). The paper would benefit from validation in additional environments.

**Questions:**

- When an option is judged infeasible, how exactly does the feasibility module generate the “alternative actions”? Is there a specific prompting strategy or rule to ensure these replacements remain consistent and task-relevant?

- Could the authors clarify how the weights in Eq. (6) are determined?

---

> ### Author Response · Authors · 2025-12-03
>
> Thank you for reviewing our paper!
> Please find below our response to the Weaknesses (W) and Questions (Q).
>
> **(W1) Presentation - Figure 1**
> Thank you for the suggestions on Figure 1. We have incorporated them to improve it in our revision.
> The revised Figure 1 annotates each step in the tree-search process.
>
> **(W2) Presentation - The results**
> As suggested, our revision includes summary tables of standard metrics in the main body.
> Please refer to our official comment titled “Summary of Revisions" for details.
>
> **(W3) Additional environments**
> During rebuttal, we further compared our ToO with the two baselines (CoT, ToT-MCTS) on 4 additional Minecraft environments with different characteristics. Please refer to our official comment titled “Summary of Revisions" for details.
>
>
> We agree that extending to non-Minecraft environments would further demonstrate the merit of ToO. We would consider it as an important follow-up work.
>
> **(Q1) Generating feasible options**
> Thank you for raising this point.
> Once an option is considered infeasible, the feasibility module (LLM_feasibility) uses a two-stage prompting process to modify the option. First, it examines the current state and identifies the specific reason for infeasibility, such as the absence of required tools, and returns a structured explanation. This explanation is then passed to a second prompting template, which directs the LLM to output a concrete, single-step option that satisfies the missing precondition while preserving the high-level intent of the original option. For example, if “Craft a bucket” is infeasible due to insufficient iron ingots, the replacement becomes “Mine 3 iron ore”. If “Milk a cow” is infeasible because no cow is nearby, the replacement becomes “Find a cow”. This design ensures that the substituted options remain executable and aligned with the intended task semantics.
>
> **(Q2) The weights in Eq. (6)**
> Thank you for raising this point.
> The weights are treated as tunable hyperparameters and, within a few iterations, we identified a set of weights that is effective for all the environments in our experiments.
>
> Specifically, to comprehensively reflect the quality of an option in task execution, we assign a higher weight of 1.2 to reward, reachability, relativity, and consistency. These terms jointly capture whether an option contributes to goal progress, whether it is executable under the current state, and whether it aligns with the historical task context. Prioritizing these indicators ensures that the scoring function favors options that advance long-horizon execution without skipping necessary intermediate steps. We also include feasibility and redundancy with moderate weights of 1.0. The feasibility weight discourages clearly infeasible options, while the redundancy weight penalizes unnecessary, repeated, or excessive options that may waste attempts on code generation during execution. Importantly, the same set of coefficients remains unchanged across all tasks to avoid task-specific tuning and maintain a stable reward structure throughout the experiments.

---

### Author Response · Authors · 2025-12-03
**Summary of Revisions**

During the rebuttal period, we incorporated the reviewers’ suggestions to revise the manuscript, with the main purpose of improving the presentation and strengthening the results:
- Per reviewer VxES, we’ve revised Figure 1 by annotating each step in the tree-search process.
- Per reviewer VxES and t5aT, we’ve included results on 4 additional Minecraft environments that are qualitatively different from the ones in the original submission, including the most difficult one (the Diamond task) suggested by t5aT. The new results consistently show the superiority of our method.
- Per several reviewers, we’ve moved most of “visualization” results in the original submission to the appendix and added in Table 1 standard metrics such as success rate, #API calls, wall-clock time etc, including both the old and new environments. We still keep a small amount of visualizations as we think they are indeed valuable for illustration and explanation purposes.
- Per several reviewers, we’ve added in Section 4.3 results for cost and scalability analysis. Specifically, in Table 2 therein, we allow all methods to spend roughly the same amount of #tokens during planning and execution and report the “pass@B” metrics, i.e., the success rate when every method has spent at most B tokens. We did this for all environments in Table 2. In Figure 4, we perform a planning scalability analysis where we plot success rate vs #tokens for the ToT-MCTS baseline and our ToO, where both share similar MCTS procedure/hyperparameters for options planning and we vary the MCTS hyperparameters to scale #tokens. The results consistently show the superiority of our ToO over ToT, as ToT-MCTS struggles to reach high success rate even when #tokens is large.

---

### Meta-Review · Area_Chair_pJPr · 2026-01-08

**Summary:**

The reviewers' consensus was negative. The primary concerns centered on the paper's initial presentation (heavily reliant on visualization over quantitative metrics), the lack of rigorous cost/scalability analysis, heavy dependence on hand-engineered prompts, and limited experimental diversity beyond similar Minecraft tasks.

**Reviewer Concerns:**

**Concerns addressed by rebuttal**
- Reviewer t5aT specifically requested the Diamond task to prove capability on harder long-horizon problems. The authors added results for this and 4 other environments.
- Reviewers HKvo and t5aT raised concerns about the computational cost of nested LLM calls. The authors provided a cost analysis and scalability plots in the revision.


**Still outstanding concerns**
- Reviewers yJ7s and HKvo pointed out the brittleness of the approach due to extensive hand-engineered prompts (for feasibility, reward prediction, etc.). While the authors argued this is standard practice, the concern regarding the method's robustness and generalization to non-Minecraft domains remains significant.
- Reviewer t5aT viewed the contribution as limited, seeing it as a variant of LATS using Voyager's skill library, questioning if the distinction between "options" and "general queries" is fundamental enough.
- Despite the added cost analysis, the fundamental issue raised by HKvo regarding the high expense of multiple nested LLM calls per MCTS iteration remains an intrinsic drawback of the architecture.

**Reviewer Scores:**

The authors made a significant effort during the rebuttal, but the initial submission had substantial gaps in rigorous evaluation. Consequently, these fundamental concerns likely limit significant improvement in the reviewers' final scores.

---

### Decision · Program_Chairs · 2026-01-26

Reject